# Tissue-specific and functional loci analysis of *CASP14* gene in the sheep horn

Xiaoning Lu[1,2], Guoqing Zhang[2], Hao Yang[2], Mingzhu Shan[2], Xiaoxu Zhang[2], Yuqin Wang[1], Junyan Bai[1]\*, Zhangyuan Pan[2]\*

1 College of Animal Science and Technology, Henan University of Science and Technology, Luoyang, China,
2 State Key Laboratory of Animal Biotech Breeding, Institute of Animal Science, Chinese Academy of Agricultural Sciences, Beijing, China

\* junyanbai@163.com (JB); zhypan01@163.com (ZP)

## Abstract

Under the current context of intensive farming, small-horned animals are more suitable for large-scale breeding. The *CASP14* gene, closely associated with skin and keratin formation, may influence horn size due to its link with skin development. This study comprehensively analyzed the tissue-specific expression of *CASP14* using RNA-Seq data, identified functional sites through whole-genome sequencing (WGS), and investigated allele-specific expression (ASE) validated by KASP assays. Results showed significantly higher *CASP14* expression in the scurred group com-pared to the SHE group, with pronounced expression in the skin. Interbreed comparisons also revealed elevated *CASP14* levels in the scurred group. Analysis of potential functional sites indicated structural similarities in the *CASP14* protein among horned mammals. Five ASE events were discovered, and intersecting these with SNPs and high fixation index loci (Fst > 0.05) identified three potential functional sites: 7941628, 7941817, and 7941830. The SNP site 7944295 was selected for T-test analysis and further validated through KASP assays, corroborating the role of *CASP14* in sheep horn phenotypes. Our findings suggest that *CASP14* plays a significant role in horn develop-ment, offering insights into breeding strategies for small-horned animals.

## Introduction

Sheep is a significant livestock species, playing a crucial role in the development of the national economy and the supply of meat [1]. To enhance the production efficiency and productivity of sheep through genetic improvement techniques, understanding traits such as growth, body weight, carcass quality, wool quality, and horn type is essential [2, 3]. Animal horns are considered cranial appendages, predominantly found among ruminant species [4]. Horns are distinctive cranial appendages of the *Bovidae* family, which includes species such as *Bos taurus (cattle)*, *Ovis aries (sheep)*, and *Capra hircus (goat)* [5]. To safeguard animals and producers from accidental injuries, polled animals better meet the requirements of current intensive livestock management practices [6–8]. Therefore, sheep horns serve as an important model for the study of phenotypic genetic evolution in animals [9]. However, the majority of current

**Data Availability Statement:** All relevant data are available from the National Center for Biotechnology Information (NCBI) database. The dataset can be accessed at the following link:

https://www.ncbi.nlm.nih.gov/sra?term=PRJNA1003277&cmd=DetailsSearch.

**Funding:** This work was supported by the National Key R&D Program Young Scientist (2023YFF1001800), National Key R&D Program of China (2022YFF1000103), National Natural Science Foundation of China (31802031 and 31960659), Agricultural Science and Technology Innovation Program of China (CAAS-ZDRW202106andASTIP-IAS13), and China Agriculture Research System of MOF and MARA (CARS-38).

**Competing interests:** The authors have declared that no competing interests exist.

research on sheep focuses on the genetic mechanisms of their primary econimic traits [10]. The genetic mechanisms underlying sheep horn development remain unclear.

Caspases (*CASP*) are cysteine proteases that play a central role in apoptosis and may also be involved in the terminal differentiation of keratinocytes, processes that could influence horn growth [11]. Caspase-14 (*CASP14*) is part of the *CASP* family. *CASP14* is a cysteine-dependent aspartate-specific protease that is expressed during the process of epidermal differentiation, is highly expressed in the skin of mammals. In humans, this protein is specifically localized to the differentiating keratinocytes, suggesting its critical involvement in the development of the epidermal barrier [12]. Furthermore, the expression of *CASP14* mRNA is confined to the uppermost viable layers of the epidermis, encompassing the granular layer, hair follicles, and sebaceous glands. This distribution implies that *CASP14* may have broader functions beyond its role as a proapoptotic gene [13]. Studies on BeWo cell lines, a choriocarcinoma-derived model for trophoblasts, have revealed a contrasting role of *CASP14* compared to its function in human epidermis. Inhibition of *CASP14* in these cells resulted in the upregulation of KLF4, hCG, and cytokeratin-18—markers associated with normal trophoblast differentiation. These findings suggest that *CASP14* influences the differentiation pathway of trophoblasts [14, 15]. Another research has indicated that the *CASP14* gene exhibits variations associated with skin immune responses and diseases, as well as variations related to maintaining skin immune homeostasis following chemical exposure [16]. Thus, as a key factor, the CASP14 gene is essential for the development of the skin barrier in animals.

Base on the previous research of our team, we have identified differentially expressed genes between large and small sheep horns and conducted Gene Ontology (GO) enrichment analysis. The results suggest that the *CASP14* gene is involved in processes such as skin barrier formation. Horns are derivatives of the skin and represent an independent organ, the development of the stratum corneum is believed to be primarily regulated by the skin [17], we speculate that the *CASP14* gene also plays a significant role in the formation of sheep horns. To further validate this hypothesis, we initially used RNA-sequencing (RNA-Seq) data to explore the tissue-specific expression of the *CASP14* gene to understand its function. Subsequently, we employed whole-genome sequencing (WGS) data to analyze functional loci within the *CASP14* gene associated with sheep horns. This study aims to investigate the molecular mechanisms underlying the formation of sheep horns, providing valuable molecular markers for sheep horn breeding.

## Materials and methods

### Ethics

Institutional Review Board Statement: All the experimental procedures mentioned in the present study were approved by the Science Research Department (in charge of animal welfare issues) of the Institute of Animal Sciences, Chinese Academy of Agricultural Sciences (IAS-CAAS) (Beijing, China). Ethical approval on animal survival was given by the animal ethics committee of IAS-CAAS (No. IASCAAS-AE-03, 12 December 2016).

### Sample collection

We collected totaling fifteen Tibetan sheep samples from Dangxiong, Tibet, China (S1 Table). All individuals in this study are female, with ages ranging from 2 to 4.5 years (PRJNA1003277) [18]. The Tibetan sheep were categorized into two groups based on horn characteristics: one with 7 sheep having scurred horns (0–12 cm), and another with 8 sheep with SHE horns (>12 cm). From the 15 sheep, we collected tissue with soft horns, placed them in deep cryopreservation tubes, and stored the tubes in liquid nitrogen.

**RNA sequencing data processing.** We performed RNA-Seq on soft horn samples from the 15 collected sheep. So our data was sourced from publicly available and laboratory collections. A total of 2,915 high-quality RNA-Seq datasets were accessed from the National Center for Biotechnology Information (NCBI) and the European Bioinformatics Institute (EBI), and combined with RNA-Seq data of our 15 sheep for the following procedures. Processing and removal of low-quality bases and artifact sequences were implemented using the TrimGalore (v.0.6.7). The clean reads were aligned to the sheep reference genome ARS-UI_Ramb_v2.0, using the STAR (v.2.5.4) with the "—chimSegmentMin 10" and "—outFilterMismatchNmax 3" parameters. High-quality RNA-Seq clean datasets were acquired for subsequent analysis, characterized by unique mapping reads exceeding 85% and a count of clean reads greater than 20,000,000. Furthermore, gene expression levels were standardized using StringTie (v.2.1.5) by calculating fragments per kilobase of transcript per million mapped reads (FPKM) and transcripts per million (TPM). Ultimately, the featureCounts (v.2.0.1) was utilized to extract the raw counts of genes.

## Analysis of *CASP14* gene expression

To explore the difference in *CASP14* gene expression between scurred and SHE groups, a boxplot was generated using the ggplot2 package (v.3.4.4) in R (v.4.3.0). To further analyze the distinctions in *CASP14* exon expression between these two groups, the genome annotation (GTF) file was formatted using the dex-seq_prepare_annotation2.py script from the Subread_to_-DEXSeq package (v.1.46.0). The formatted GTF file and the counts matrix produced by featureCounts were pro-cessed using load_SubreadOutput.R in RStudio, enabling the construction of the DEXSeqDataSetFromFeatureCounts (dds) object. Subsequently, an exon difference analysis was conducted to identify differential exon expression between the groups.

We obtained RNA-Seq data from the Genotype-Tissue Expression (GTEx) project, encompassing a total of 2651 pig samples, 4359 cow samples, and 9810 human samples, along with their TPM values (S2 Table). RNA-Seq data from sheep, pigs, cows, and humans were aggregated and utilized to categorize samples into 16 tissue types, subsequently computed the average TPM values for each species. Moreover, data on eight sheep breeds were extracted from public RNA-Seq databases, the selected breeds were Carpet, Rambouillet, Tibetan, Bashibai, Chinese Merino, Hu, Spanish Churra, Tan, which were categorized into scurred, and SHE subgroups.

## Evolutionary and structural analysis of *CASP14*

We retrieved the FASTA files for the gene of interest in 20 different species from the NCBI database and constructed a phylogenetic tree for the *CASP14* gene using the online tool iTOL (https://itol.embl.de/). To ascertain whether the amino acids are located at key positions, we predicted the 3D structure of the *CASP14* protein using AlphaFold2 (https://colab.research.google.com/github/sokrypton/ColabFold/blob/main/AlphaFold2.ipynb), which generated five models. From these, we selected the model with the highest score.

## Analysis and validation of whole-genome sequencing

The sequencing data sets consisted of WGS generated in this study and published previously, including 3125 sheep from PRJNA304478, PRJNA325682, PRJNA479525, PRJNA624020, PRJNA675420, PRJNA822017, PRJNA30931, PRJNA480684, PRJNA509694, PRJNA779188, and PRJNA783661 [19–30]. A sample of 3,125 sheep was classified into scurred and SHE groups. F-statistic (Fst) values were calculated using VCFtools (v.0.1.16) to identify SNP loci exhibiting significant differences between these two populations. Fst value is a measure of

genetic differentiation between populations, with higher values indicating greater genetic divergence. Small Tail Han sheep were selected as the experimental breed, and horn length was quantified in 36 individuals by measuring from the horn base to its outermost end. The right horn length was used to determine the overall horn size. Sequencing reads were trimmed using Trimmomatic (v.0.39), and the raw sequencing data quality was assessed with FastQC (v.0.12.1). The qualified reads were then aligned, sorted, and merged to the sheep reference genome using BWA (v.0.7.17) and Picard (v.3.1.1). Mutation information was annotated using SnpEff (v.4.3). A dominant model was applied to establish the relationship between horn length and different genotypes. For further analysis of the SNP chain, the VCFtools (v.0.1.16) filters—maf 0.45 and—min-meanDP 5 were used, and LDBlockShow (v.1.40) was employed to identify SNP loci (S3 Table).

Finally, the identified loci were sent to a biotechnology company for validation through KASP (Kompetitive Allele-Specific PCR) assays.

## Results and discussion

### Comparative *CASP14* expression across sheep horn phenotypes

We analyzed CASP14 gene expression in different horn types of sheep and various tissues. The analysis revealed that CASP14 gene was significantly upregulated in the scurred group compared to the SHE group (p = 0.019), indicating a potential role in scurred horn development (Fig 1A). Further analysis of the CASP14 gene exon-specific expression showed consistently higher expression in the scurred group (Fig 1B). All exons of the *CASP14* gene were expressed in both the scurred and SHE groups (Fig 1C). Additionally, the *CASP14* gene exhibited relatively high expression in the skin, while its expression levels were low or even undetectable in the blood, central nervous system (CNS), heart, liver, lung and muscle. These findings suggested that the CASP14 gene is closely linked to sheep with scurred horn type.

### *CASP14* tissue specificity in sheep

We selected different species to further investigate the difference of the CASP14 expression. We found that the CASP14 gene was most highly expressed in human skin, followed by sheep skin and cattle skin (Fig 2A). To explore potential sex-related differences in CASP14 expression across sheep tissues, we analyzed gene expression in male and female sheep (Fig 2B). *CASP14* gene expression in skin tissue was elevated in both female and male subjects relative to other tissues.Furthermore, the expression of the *CASP14* gene was examined across different breeds (Fig 2C). Within the SHE group (Carpet, Rambouillet, Tibetan), *CASP14* gene expression was the lowestwhile higher expression were noted in scurred sheep breeds (Bashibai, Chinese Merino, Hu, Spanish Churra, Tan). These analyses further highlighted the significance of CASP14 gene expression in skin and scurred phenotype.

### The potential functions of the *CASP14* gene

We then conducted a comprehensive analysis of the amino acid sequence and protein structure of CASP14. The phylogenetic analysis of the *CASP14* gene was consistent with the evolutionary relationships among species, showing that *Bovidae* and *Cervidae* cluster together (Fig 3A). Similarly, there were amino acid sites unique to horned animals, with positions 11, 146, 148, 152, 153, 169, 172, 198, and 218 exhibiting a consistent pattern (Fig 3C). These results indicated structural similarities in the CASP14 protein among these horned mammals. These conserved residues may play a crucial role in the growth and development of sheep horns. Furthermore, the protein structure of *CASP14* revealed that the amino acid at position 218

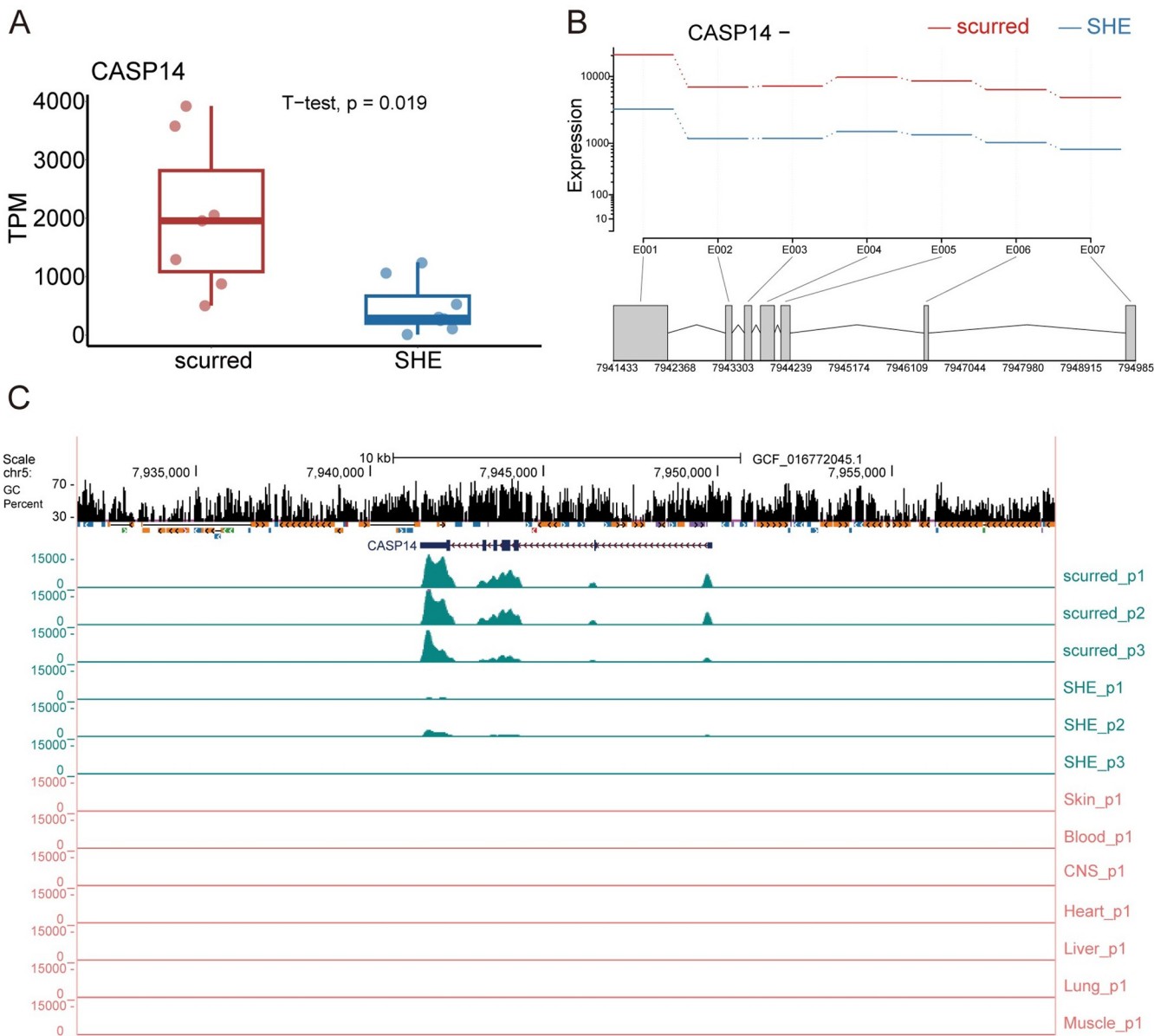

**Fig 1. Expression of *CASP14* gene in scurred and SHE groups.** (A) Differential expression of *CASP14* gene between scurred and SHE Groups. (B) Expression of *CASP14* exons in scurred and SHE groups. The expression refers to the estimated values derived from the fitted expressions in the GLM regression, where E001 to E007 represent the number of exons. (C) Differential expression of *CASP14* in horn and other tissues.

(Fig 3B), situated within a beta-sheet region, was consistently methionine in horned animals such as sheep, suggesting a significant association with horn presence.

## Characterization of allele-specific expression in *CASP14* gene

Allele-specific expression (ASE) is a phenomenon in which one allele is preferentially expressed over the other, potentially influencing phenotypic traits. In our study, analysis of RNA-Seq data revealed five allele-specific expressions (ASEs): ASE1 (Chr5: 7941514), ASE2 (Chr5: 7941569), ASE3 (Chr5: 7941628), ASE4 (Chr5: 7941817), and ASE5 (Chr5: 7941830),

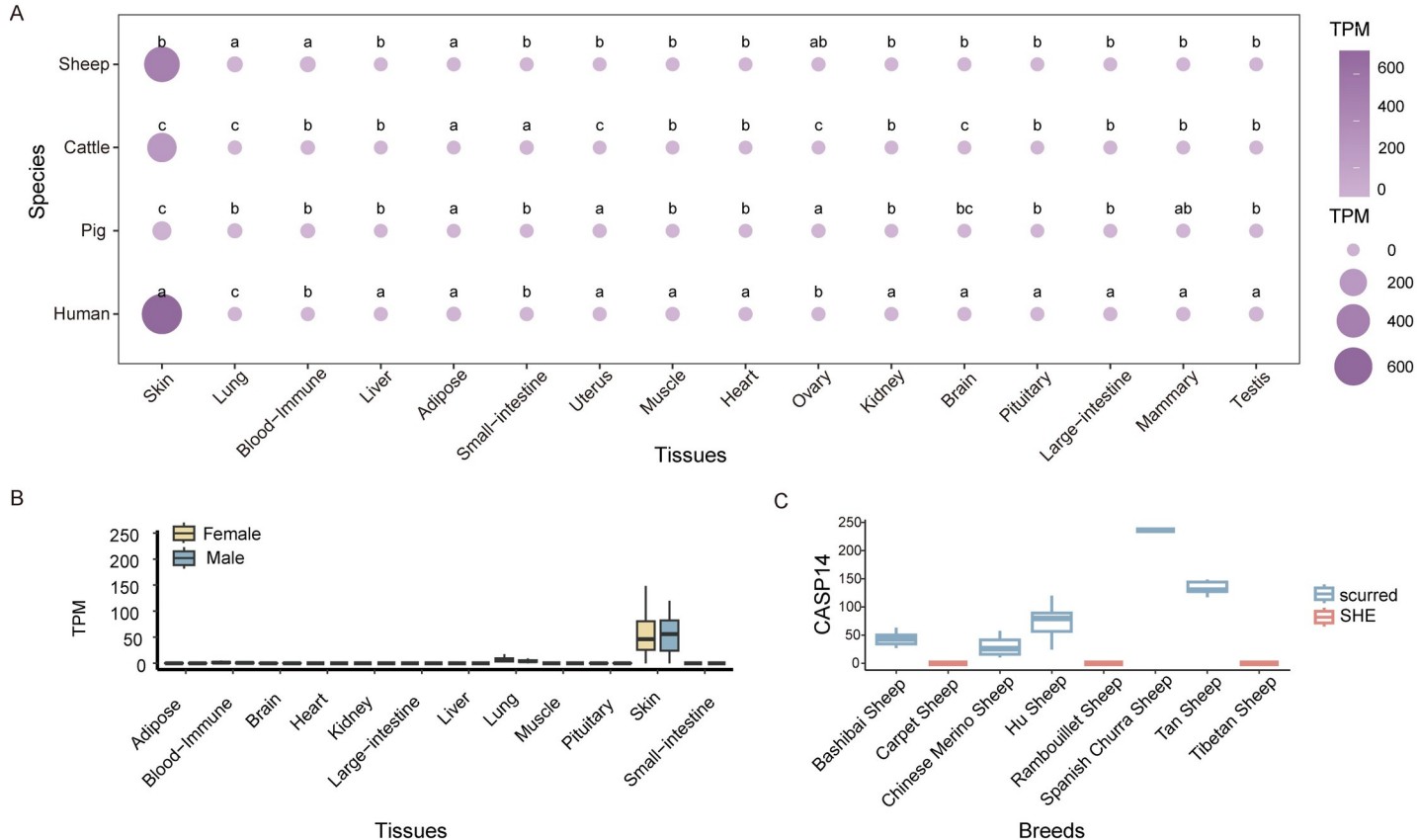

**Fig 2. Expression differences of *CASP14* gene.** (A) Differential expression of *CASP14* gene in four species. (B) Sex differences in *CASP14* gene expression across various sheep tissues. (C) Ex-pression of *CASP14* gene in various sheep breeds.

all situated within Exon 7 (Fig 4). These ASE locis we found might be closely linked to sheep horn types.

## Potential functional mutations in the *CASP14* gene

To observe the phenotypic differences between the scurred group and the SHE group under the influence of the *CASP14* gene, we conducted a principal component analysis (PCA). The results demonstrated a clear separation between scurred and SHE sheep breeds (Fig 5A). And we identified several locis with Fst values greater than 0.05 (Fig 5B), some of which were located within the exon regions of the *CASP14* gene. These locis had significant implications for studying the functional and adaptive differences of the *CASP14* gene between the scurred and the SHE populations. As presented in Table 1, these were some SNP sites of the *CASP14* gene, combined with the analysis of Fst values, these sites may be closely associated with the size of sheep horns.

Further analysis was conducted to investigate the key regulatory sites involved in CASP14 gene expression. The analysis showed the intersection of single nucleotide polymorphisms (SNPs), allele-specific expressions (ASEs), and Fst at the loci 7941628, 7941817, and 7941830 (Fig 6A). These loci likely play important roles in genetic diversity, population structure, and gene expression regulation. They may represent key genes or regulatory regions under selec-tive pressure that influence allele-specific expression. Moreover, we found the complex genetic

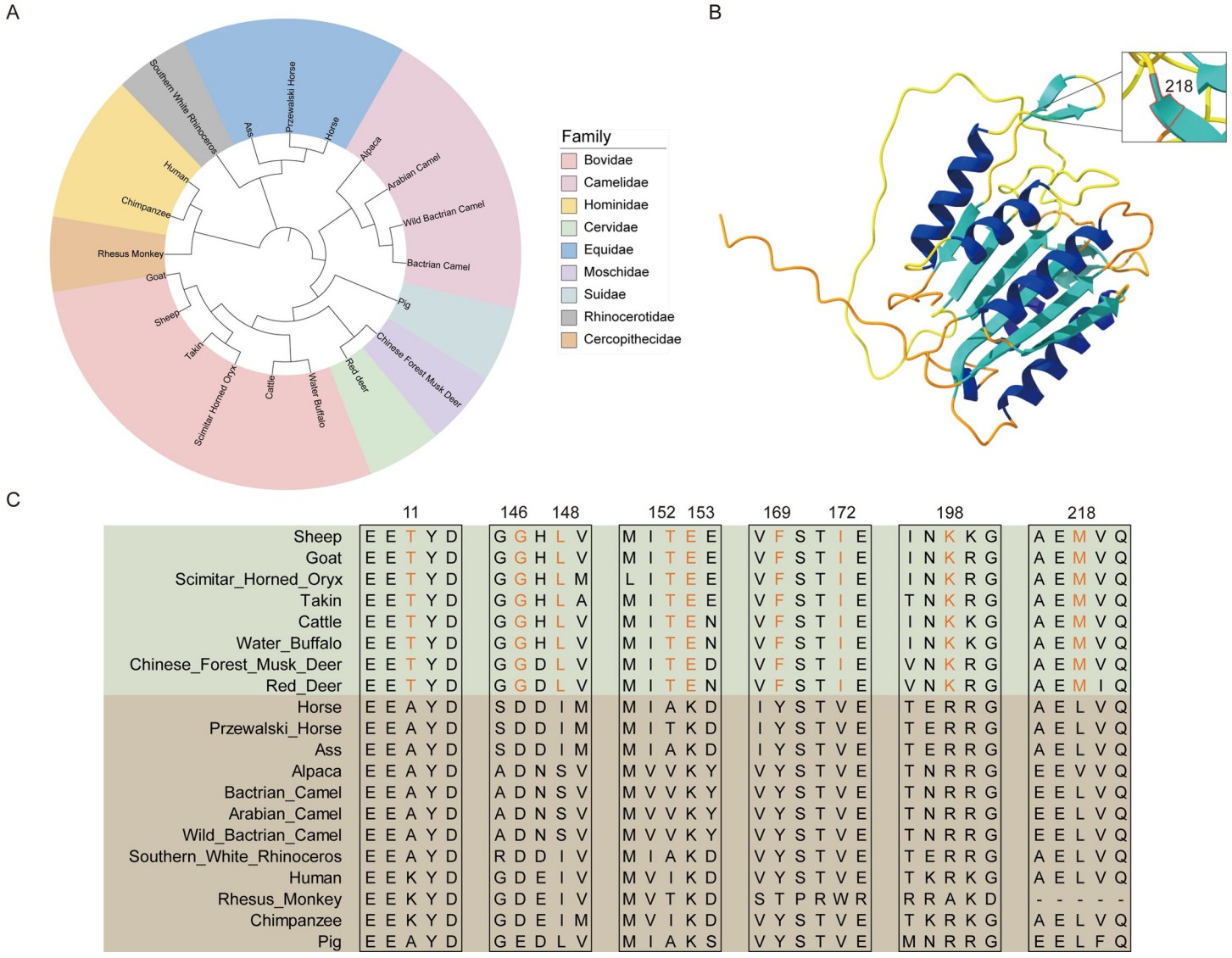

**Fig 3. Structural and evolutionary insights of the *CASP14* gene.** (A) Evolutionary tree of the *CASP14* gene. (B) Three-Dimensional (3D) structure of sheep *CASP14* protein. (C) Amino acid sites of *CASP14* gene unique to horned animals.

associations among SNP loci within the 7.942 Mb to 7.950 Mb region on chromosome 5. Multiple high linkage disequilibrium (LD) blocks have been identified, indicating potential regions of functional importance where SNPs exhibited significant genetic linkage. The deep orange areas and SNPs within the black triangular regions, representing high LD, were focal points for further research and may contain genes related to important traits or diseases.

## Validation of SNP loci using KASP assay

We selected the specific SNP loci chr5:7944295, there was a significant disparity in horn length among sheep with different genotypes (Fig 7A). This SNP site was likely to influence the size of the sheep horns. To validate the identified SNP site, we employed KASP technology. Through rigorous experimental procedures, we successfully genotyped the selected SNP sites in our sample population (Fig 7B).

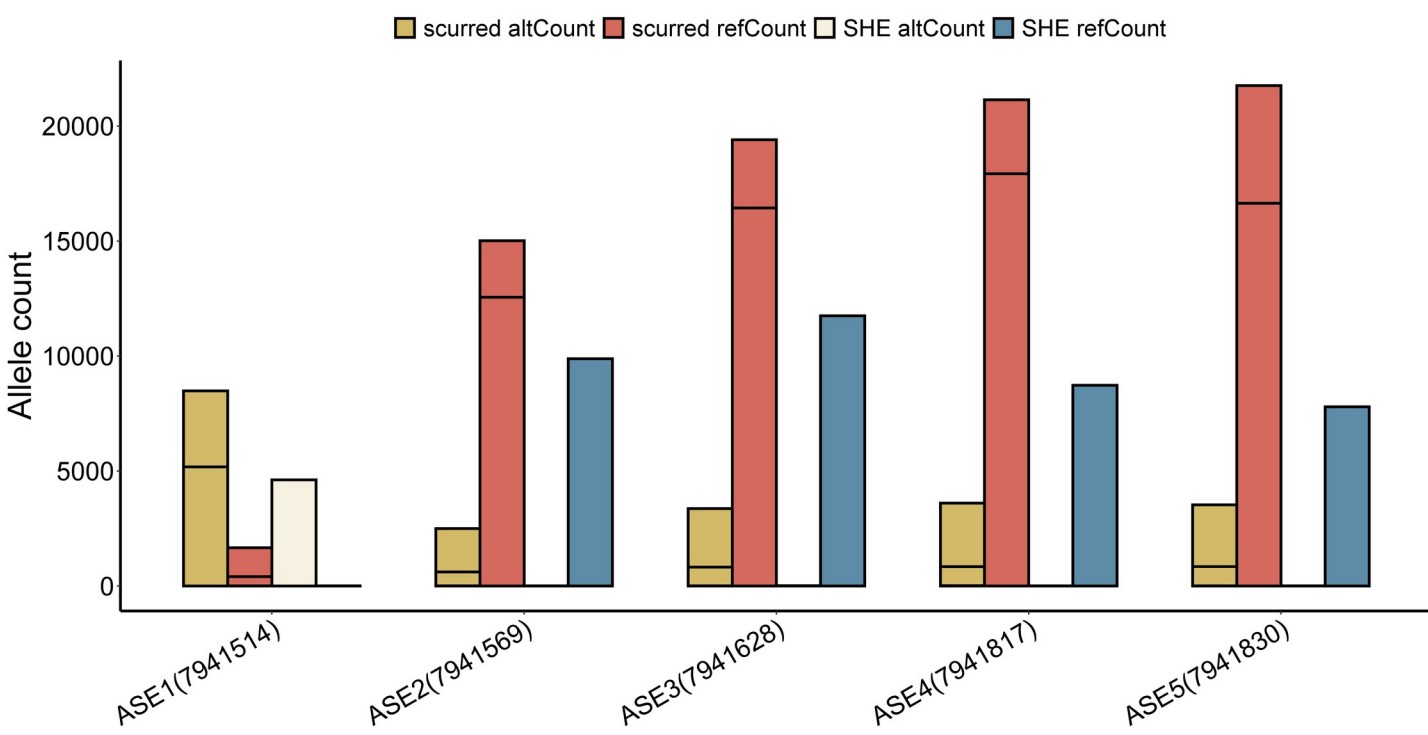

**Fig 4. Allele count of 5 ASEs differing between scurred and SHE groups.**

## Discussion

Previous studies have identified genes such as RXFP2, INSL3, KRT10, and WNT7B as being associated with horn growth [31, 32]. For the first time, researchers utilized CRISPR/Cas9 technology to successfully create a cryptorchid sheep model by targeting the RXFP2 gene. However, the findings of this study indicated that partial disruption of the RXFP2 gene did not affect the horn size in sheep, suggesting that partial disruption of RXFP2 was associated with testicular descent rather than horn formation [33]. This indicated that the gene regulating the sheep horn phenotype was not singular, as demonstrated by the RXFP2 disruption not affecting horn size, thus necessitating a more in-depth investigation into the genetic mechanisms underlying the sheep horn phenotype.

*CASP14* belongs to a distinct group within the highly conserved cysteine aspartate-specific protease family [34]. *CASP14* is activated at the junction between the granular and cornified layers, with both its active and inactive forms observable in epidermal extracts, while only the active form is present in the cornified layer [35]. This activation coincides with stratum corneum formation during embryonic development [36, 37]. Our findings indicate that the *CASP14* gene is significantly upregulated in the scurred group compared to the SHE group. Additionally, the *CASP14* gene is highly expressed in the skin, exhibiting higher expression levels in both humans and sheep. The results are consistent with existing research findings [38, 39]. Our study revealed that mutations in the *CASP14* gene affect the development of skin derivatives-horn. By analyzing allele-specific expression sites (ASE), we identified five key loci within the *CASP14* gene region that may influence horn size variation. Additionally, significant differences were observed between horned and polled sheep at the screened single nucleotide polymorphism (SNP) loci. Using a dominant model and whole-genome sequencing (WGS) data, we determined that the variant locus g.7944295G>A is significantly associated

A

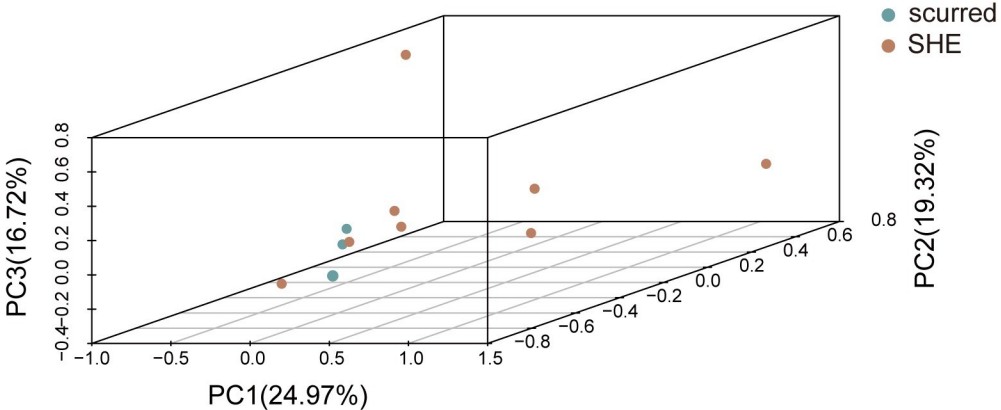

B

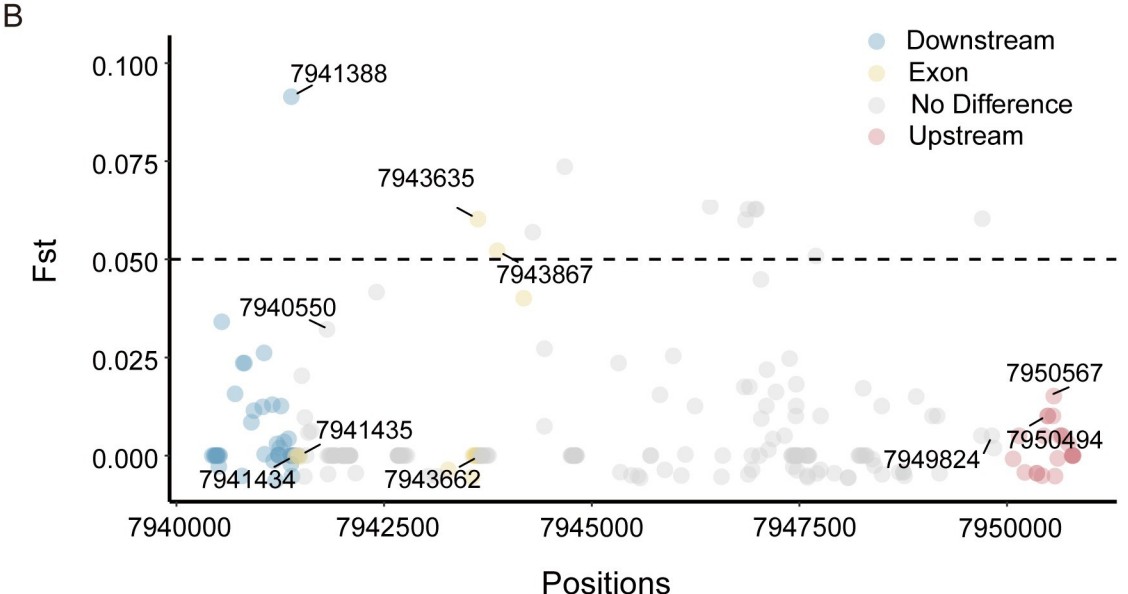

**Fig 5. Genetic variation and population structure of the *CASP14* gene.** (A) Three-dimensional (3D) principal component analysis (PCA) of the *CASP14* gene. Each point represents a sheep breed, which is categorized into scurred and SHE types based on breed information. (B) F-statistics (Fst) of functional loci of the *CASP14* gene.

with horn length in sheep. This variant likely plays a crucial role in horn growth, highlighting the impact of *CASP14* gene mutations on the morphology of skin derivatives. CK18 was expressed in normal epithelial cells of most organs but absent in normal squamous epithelium [40]. A study indicates that inhibition of *CASP14* in Be-Wo cells resulted in the upregulation of KLF4, hCG, and CK18-markers associated with normal trophoblast differentiation. These findings suggest that *CASP14* influences the differentiation pathway of trophoblasts. Our study demonstrates that the expression levels of *CASP14* are elevated in the scurred group, corroborating previous research on the influence of *CASP14* on keratinization processes.

**Table 1. Potential SNP functional loci of *CASP14* gene.**

| INFO | Mutation Type | Position | Fst | Ref | Alt |
|---|---|---|---|---|---|
| SNP1 | downstream | 7940550 | 0.0340909 | A | C |
| SNP2 | downstream | 7941388 | 0.0914412 | G | A |
| SNP3 | missense | 7943635 | 0.0602477 | T | C |
| SNP4 | intron | 7944295 | 0.0569244 | G | A |
| SNP5 | intron | 7946417 | 0.101679 | C | T |
| SNP6 | intron | 7946876 | 0.0784427 | G | A |
| SNP7 | exon | 7947041 | 0.0448705 | G | A |
| SNP8 | exon | 7947467 | 0.0181818 | C | T |
| SNP9 | upstream | 7950567 | 0.0151515 | C | T |
| SNP10 | upstream | 7950494 | 0.010101 | G | A |

The mechanisms governing CASP14 gene transcription remain incompletely understood, with expression in keratinocytes occurring only under specific conditions like high-density culture, forced aggregation, or vitamin D3-induced differentiation [34, 38, 39, 41]. Sphingolipid metabolites, particularly ceramides, are key regulators in cellular processes and have been identified as inducers of *CASP14* [42]. The up-regulation of *CASP14* by these lipids is mediated through the inhibition of kinase pathways, highlighting its role in cell differentiation and apoptosis [43]. Which may also impact the development of the horn phenotype. Treatment of human keratinocytes with Th2 cytokines IL-4 and IL-13 significantly reduces *CASP14* protein expression [44]. The *CASP14* gene is expressed in the differentiated stratum corneum and hair follicles of the epidermis, with its distribution in the epidermis and hair follicles being maintained across multiple mammalian species as revealed by ultrastructural analysis [45, 46]. *CASP14* is involved in the formation of the embryonic skin barrier, with its expression

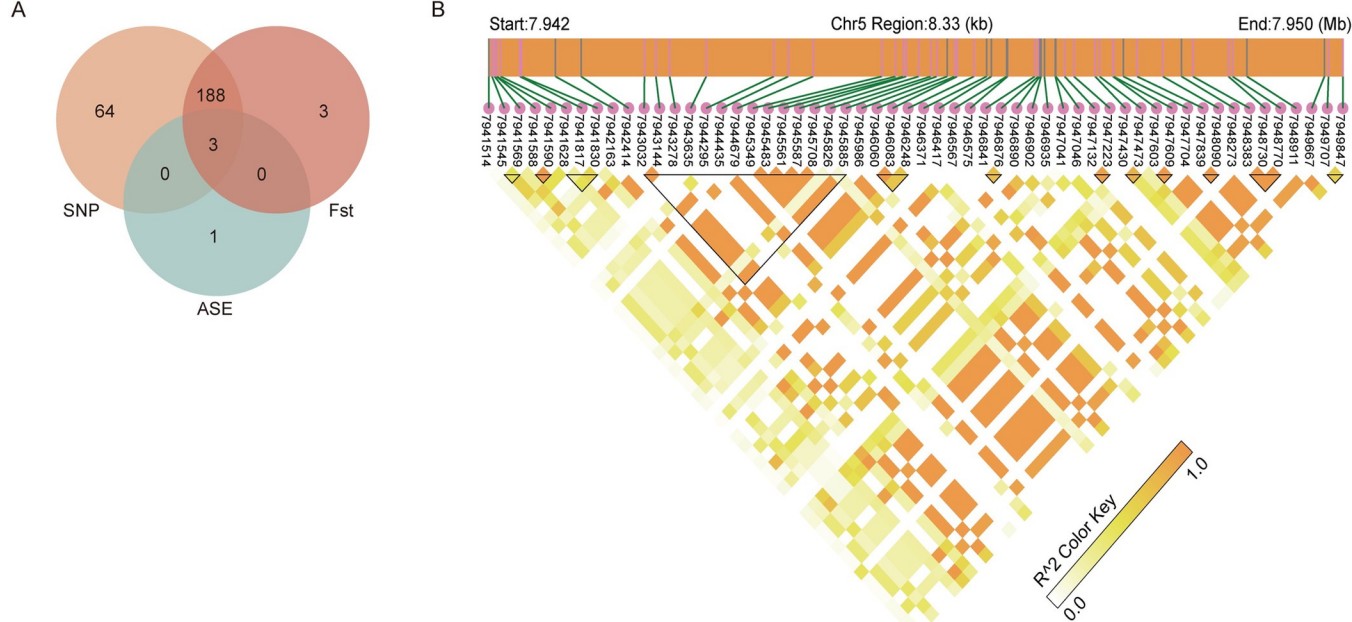

**Fig 6. Genomic insights into the *CASP14* gene.** (A) Intersections of ASEs, SNPs and Fst. (B) LD heatmap of *CASP14* gene. Darker colors represent higher LD values, and black triangles denote LD blocks, which are collections of SNPs with higher LD values.

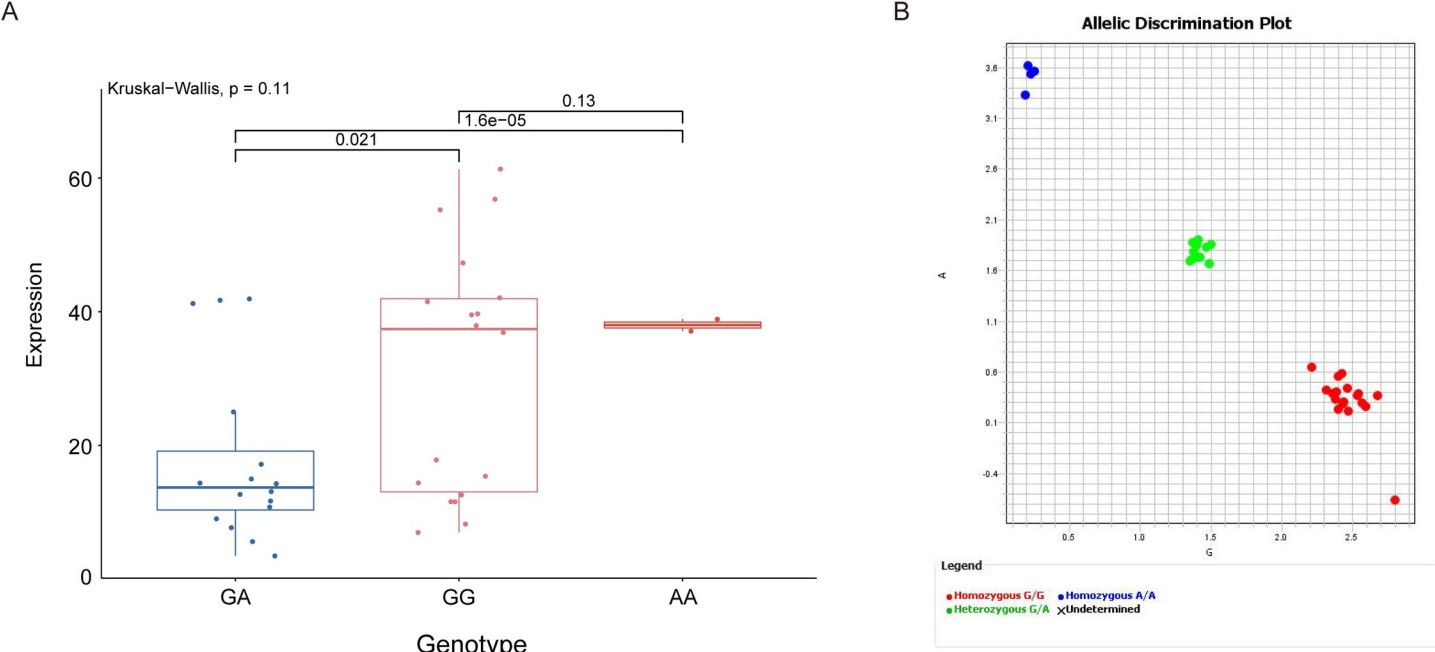

**Fig 7. Phenotypic and genotypic analysis of the *CASP14* gene.** (A) Box plot of individual horn lengths for different genotypes. Calculating the p-value based on the dominant genetic model and t-test. (B) The KASP assay results for the *CASP14* gene.

initiating from embryonic day 14.5 or 15.5 and processing commencing from day 17.5, occurring alongside the formation of the epidermal stratum corneum [35, 47]. The predominant location of *CASP14* in the epidermis suggests it may play a specialized role there, the deepest layer of the epidermis, known as the basal layer, contains actively dividing keratinocytes. [48]. In this study, an investigation of the expression patterns of *CASP14* across various tissues revealed that the gene is highly expressed in the skin, whereas its expression in other tissues such as the heart, blood, and CNS is relatively low or nearly absent. Furthermore, in the skin, *CASP14* gene expression was significantly higher than in other tissues across four species. These results are similar to those of the aforementioned studies. The growth of horns is closely related to the skin, and the expression patterns of the *CASP14* protein in horned animals such as sheep, cattle, and deer within the Artiodactyla order exhibit similar trends, indicating a close association of the *CASP14* gene with horn growth.

## Conclusion

The study demonstrated a correlation between the *CASP14* gene and the size of sheep horns. Initial analysis showed CASP14 gene expression is higher in the scurred group. Further analysis indicated that the *CASP14* gene was highly expressed in the skin tissues of both rams and ewes. We conducted an in-depth analysis of the amino acid sequence sites and protein structure of the *CASP14* gene, identifying nine amino acid positions that were specifically expressed in horned animals. Additionally, by integrating the analysis of SNPs, ASEs, and Fst of the *CASP14* gene, we identified several potential functional sites of the gene, with the SNP g.7944295 G > A potentially having a significant impact on the size of sheep horns. These sites may significantly affect the size and type of sheep horns and can serve as molecular markers for horn breeding.

## Supporting information

**S1 Table. Sample infromation.**
(XLSX)

**S2 Table. The mean TPM of the *CASP14* gene in various tissues of cattle, humans, pigs, and sheep.**
(XLSX)

**S3 Table. SNP information of *CASP14* gene.**
(XLSX)

## Author Contributions

**Data curation:** Xiaoning Lu, Guoqing Zhang.

**Investigation:** Junyan Bai.

**Methodology:** Xiaoning Lu, Guoqing Zhang, Xiaoxu Zhang.

**Software:** Hao Yang.

**Supervision:** Yuqin Wang, Junyan Bai.

**Validation:** Mingzhu Shan.

**Visualization:** Xiaoning Lu.

**Writing – original draft:** Xiaoning Lu.

**Writing – review & editing:** Zhangyuan Pan.

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
