## [Decision Letter · Decision Letter 0]

25 Sep 2024

PONE-D-24-36938Tissue-specific and functional loci analysis of CASP14 gene in the sheep hornPLOS ONE

Dear Dr. Pan,

Thank you for submitting your manuscript to PLOS ONE. After careful consideration, we feel that it has merit fully meet PLOS ONE’s publication criteria as it currently stands. Therefore, we invite you to submit a revised version of the manuscript that addresses the points raised during the review process.

We look forward to receiving your revised manuscript.

Kind regards,

Sayed Haidar Abbas Raza

Academic Editor

PLOS ONE

Journal Requirements:

plos.org/plosone/s/file?id=wjVg\\PLOSOne_formatting_sample_main_body.pdf%20 and When submitting your revision, we need you to address these additional requirements.

https://doi.org/10.3390/genes15030376

https://doi.org/10.3390/ijms22115575

In your revision ensure you cite all your sources (including your own works), and quote or rephrase any duplicated text outside the methods section. Further consideration is dependent on these concerns being addressed.

3. Thank you for stating the following financial disclosure: “This work was supported by the National Key R&D Program Young Scientist(2023YFF1001800), National Key R&D Program of China (2022YFF1000103), National Natural Science Foundation of China (31802031 and 31960659), Agricultural Science and Technology Innovation Program of China (CAAS-ZDRW202106andASTIP-IAS13), and China Agriculture Research System of MOF and MARA (CARS-38).”

4. Please note that your Data Availability Statement is currently missing the repository name. If your manuscript is accepted for publication, you will be asked to provide these details on a very short timeline. We therefore suggest that you provide this information now, though we will not hold up the peer review process if you are unable.

Reviewers' comments:

Reviewer's Responses to Questions

**Comments to the Author**

1. Is the manuscript technically sound, and do the data support the conclusions?

Reviewer #1: Yes

Reviewer #2: Yes

2. Has the statistical analysis been performed appropriately and rigorously? 

Reviewer #1: No

Reviewer #2: Yes

3. Have the authors made all data underlying the findings in their manuscript fully available?

Reviewer #1: Yes

Reviewer #2: Yes

4. Is the manuscript presented in an intelligible fashion and written in standard English?

Reviewer #1: Yes

Reviewer #2: No

5. Review Comments to the Author

Reviewer #1: The manuscript "Tissue-specific and functional loci analysis of CASP14 gene in the sheep horn" uses genomic technologies to study the CASP14 gene loci in sheep horn. It is suggested that sheep horns serve as an important model for the study of phenotypic genetic evolution in animals. CASP14 is a cysteine dependent aspartate-specific protease that is expressed during the process of epidermal differentiation, and is highly expressed in the skin of mammals. Horns are derivatives of the skin and represent an independent organ and the authors hypothesize that CASP14 gene also plays a significant role in the formation of sheep horns. To test their hypothesis they collected RNA-Seq data for 15 Tibetan sheep samples from two groups scurred and SHE. They also collected several RNA-Seq datasets from NCBI and EBI. All the data was processed through the same pipeline. They found that the expression of CASP14 gene was significantly higher in the scurred compared to the SHE. They then performed a PCA analysis and found a clear separation between scurred and SHE sheep breeds. The authors conclude by saying that this study demonstrates a correlation between the CASP14 gene and the size of sheep horns and that the CASP14 gene is highly expressed in the skin tissues of both rams and ewes.

I have several concerns with this manuscript

1. The language of the manuscript is very confusing. For example it is not clear whether the authors performed an RNA sequencing experiment on 15 sheep samples for thus study or they just retrieved data from some previously published study. The latter seems to be the case.

2. Their numbers seems to be incorrect, on line 68 they talk about 2915 RNA-seq datasets. The authors need to double check this.

3. The result section is very unclear and needs to be rewritten. For example in line 125 the results start with Fig2A. The authors need to look at some published manuscripts and write the results accordingly.

4. The figures need to be labeled properly especially Fig5B.

5. In their conclusion the authors state that there is a correlation between the CASP14 gene and the size of sheep horns but they never explained in the beginning of the manuscript which sheep has what horn size.

Reviewer #2: This research article provides a comprehensive investigation into the role of the CASP14 gene in the development of sheep horns, combining RNA-Seq, whole-genome sequencing (WGS), and bioinformatics tools to explore the gene’s expression patterns and potential genetic markers. The use of multiple species and comparative gene expression analysis adds depth to the study, making it a valuable contribution to livestock genetics and horn phenotype research.

Positive Aspects:

Multi-faceted Approach: The combination of RNA-Seq, WGS, and phylogenetic analysis offers a well-rounded exploration of CASP14 gene expression and its relation to horn size.

Significant Findings: The identification of key loci and the potential role of the SNP g.7944295 G>A in influencing horn size is a valuable outcome for sheep breeding.

Cross-Species Analysis: The inclusion of multiple species (sheep, cattle, deer) for comparative analysis strengthens the study’s broader applicability in understanding horn growth across different ruminants.

Suggestions for Minor Corrections:

Grammar and Spelling:

Line 31: "eco-nomic" should be corrected to "economic."

Line 40: "epi-dermis" should be "epidermis."

Line 70: "alongside laboratory-collected samples" is slightly unclear. It would be clearer as "in addition to laboratory-collected samples."

Line 110: "Kompetitive Allele-Specific PCR" should consistently be abbreviated as "KASP" for better flow, as the acronym is introduced later.

Line 221: "coinciding with the formation of the epidermal stratum corneum" should be rephrased for clarity, as the word "coinciding" is somewhat repetitive throughout the article.

Technical Consistency:

"disruption of RXFP2 is associated with testicular descent" lacks context regarding its relevance to horn growth. A clearer link should be made to explain why this finding is significant for the current study.

"low or even undetectable in the blood, CNS, heart, liver, lung and muscle" – the term "CNS" should be spelled out fully the first time it is used, as it may not be immediately recognized by all readers.

Flow and Clarity:

The transitions between sections discussing CASP14's role in apoptosis and its relation to horn growth could benefit from smoother transitions to connect these ideas more explicitly.

More explanation is needed when introducing the Fst analysis results. The significance of Fst values should be briefly clarified for readers unfamiliar with population genetics.

Addressing these minor grammatical and technical inconsistencies would enhance the overall clarity and readability of the article while maintaining the strength of its scientific findings.

6. PLOS authors have the option to publish the peer review history of their article (what does this mean?). If published, this will include your full peer review and any attached files.

Reviewer #1: No

Reviewer #2: **Yes: **Simna Saraswathi Prasannakumari

---

## [Author Response · Author response to Decision Letter 0]

12 Oct 2024

Reviewer #1:

1. The language of the manuscript is very confusing. For example it is not clear whether the authors performed an RNA sequencing experiment on 15 sheep samples for thus study or they just retrieved data from some previously published study. The latter seems to be the case.

Author response: Thank you for pointing this out. We apologize for not clearly stating the data, and we have added a section on line 69 of the manuscript to explain our data processing. 

Line 69: We performed RNA-Seq on soft horn samples from the 15 collected sheep.

2. Their numbers seems to be incorrect, on line 68 they talk about 2915 RNA-seq datasets. The authors need to double check this.

Author response: Thank you for catching this. Our RNA-Seq data consists of two main parts: one part is the RNA-Seq data from the 15 sheep generated in our lab, as mentioned earlier, and the other part includes 2915 collected RNA-Seq datasets from the National Center for Biotechnology Information (NCBI) database (https://www.ncbi.nlm.nih.gov/) and EBI (https://www.ebi.ac.uk/), which are publicly available. These two parts were combined for subsequent analysis. We have improved the description of datasets in lines 70-73.

Lines 70-73: A total of 2,915 high-quality RNA-Seq datasets were accessed from the National Center for Biotechnology Information (NCBI) and the European Bioinformatics Institute (EBI), and combined with RNA-Seq data of our 15 sheep for the following procedures.

3. The result section is very unclear and needs to be rewritten. For example in line 125 the results start with Fig2A. The authors need to look at some published manuscripts and write the results accordingly.

Author response: We agree with the reviewer’s assessment. We understand the importance of a clear structure in the results section. We have reviewed published manuscripts in this journal to guide our revision and have restructured all of the results section accordingly. The section now introduces findings more clearly, such as Fig 2A, have been appropriately contextualized within the text. 

Lines 132-145: We selected different species to further investigate the difference of the CASP14 expression. We found that the CASP14 gene was most highly expressed in human skin, followed by sheep skin and cattle skin (Fig 2A). To explore potential sex-related differences in CASP14 expression across sheep tissues, we analyzed gene expression in male and female sheep (Fig 2B). CASP14 gene expression in skin tissue was elevated in both female and male subjects relative to other tissues. Furthermore, the expression of the CASP14 gene was examined across different breeds (Fig 2C). Within the SHE group (Carpet, Rambouillet, Tibetan), CASP14 gene expression was the lowest while higher expression were noted in scurred sheep breeds (Bashibai, Chinese Merino, Hu, Spanish Churra, Tan). These analyses further highlighted the significance of CASP14 gene expression in skin and scurred phenotype.

4. The figures need to be labeled properly especially Fig5B.

Author response: We have thoroughly reviewed and adjusted all figure labels to ensure clarity and accuracy, with particular attention to Fig 5B. In Figure 5B, we plotted each locus based on its Fst value, categorizing the loci into four main types: Downstream, Upstream, Exon and No difference. With some locis randomly selected for labeling in the figure.

5. In their conclusion the authors state that there is a correlation between the CASP14 gene and the size of sheep horns but they never explained in the beginning of the manuscript which sheep has what horn size.

Author response: Thank you for your suggestion. In the Materials and Methods section, we specified that the 15 sheep were divided into two groups: the scurred group, with horn lengths ranging from 0-12 cm, and the SHE group, with horn lengths greater than 12 cm. Detailed horn length for each sheep have also been added to Supplementary Material 1.

Lines 63-67: We collected totaling fifteen Tibetan sheep samples from Dangxiong, Tibet, China. All individuals in this study are female, with ages ranging from 2 to 4.5 years (PRJNA1003277) [18]. The Tibetan sheep were categorized into two groups based on horn characteristics: one with 7 sheep having scurred horns (0–12 cm), and another with 8 sheep with SHE horns (>12 cm). From the 15 sheep, we collected tissue with soft horns, placed them in deep cryopreservation tubes, and stored the tubes in liquid nitrogen.

Reviewer #2:

1. Grammar and Spelling

Line 31: "eco-nomic" should be corrected to "economic."

Line 40: "epi-dermis" should be "epidermis."

Line 70: "alongside laboratory-collected samples" is slightly unclear. It would be clearer as "in addition to laboratory-collected samples."

Line 110: "Kompetitive Allele-Specific PCR" should consistently be abbreviated as "KASP" for better flow, as the acronym is introduced later.

Line 221: "coinciding with the formation of the epidermal stratum corneum" should be rephrased for clarity, as the word "coinciding" is somewhat repetitive throughout the article.

Author response: Thank you for your careful review and helpful comments regarding grammar and spelling. We have addressed each point as follows:

We have corrected “eco-nomic” to “economic” on line 31, and we have updated “epi-dermis” to “epidermis” on line 41.

In lines 70-73 of the manuscript, we have rephrased the description to improved clarity.

Line 70-73: A total of 2,915 high-quality RNA-Seq datasets were accessed from the National Center for Biotechnology Information (NCBI) and the European Bioinformatics Institute (EBI), and combined with RNA-Seq data of our 15 sheep for the following procedures.

We have standardized “Kompetitive Allele-Specific PCR” to “KASP” and introduced the abbreviation upon its first mention in lines 113-114. 

Lines 113-114: Finally, the identified loci were sent to a biotechnology company for validation through KASP (Kompetitive Allele-Specific PCR) assays.

We have revised the phrase on line 245 of the manuscript to reduce repetition, rephrasing it to improve clarity. The updated wording now reads: “CASP14 is involved in the formation of the embryonic skin barrier, with its expression initiating from embryonic day 14.5 or 15.5 and processing commencing from day 17.5, occurring alongside with the formation of the epidermal stratum corneum.”

2. Technical Consistency:

"disruption of RXFP2 is associated with testicular descent" lacks context regarding its relevance to horn growth. A clearer link should be made to explain why this finding is significant for the current study.

"low or even undetectable in the blood, CNS, heart, liver, lung and muscle" – the term "CNS" should be spelled out fully the first time it is used, as it may not be immediately recognized by all readers.

Author response: As suggested by the reviewer, we have made some adjustments to our original description to clarify our findings in lines 210-216. In addition, our study highlighted that knocking out RXFP2, a gene closely linked to horn growth, did not affect horn size, suggesting that the regulatory mechanisms of horn growth in sheep remain complex and not yet fully understood. This underscores the importance of our study in exploring the gene regulatory mechanisms underlying horn growth, which we aim to investigate further.

lines 210-216: For the first time, researchers utilized CRISPR/Cas9 technology to successfully create a cryptorchid sheep model by targeting the RXFP2 gene. However, the findings of this study indicated that partial disruption of the RXFP2 gene did not affect the horn size in sheep, suggesting that partial disruption of RXFP2 was associated with testicular descent rather than horn formation [33]. This indicated that the gene regulating the sheep horn phenotype was not singular, as demonstrated by the RXFP2 disruption not affecting horn size, thus necessitating a more in-depth investigation into the genetic mechanisms underlying the sheep horn phenotype.

On line 124: The term “CNS” is now spelled out as “central nervous system” upon its first mention to ensure clarity for all readers.

Line 124: Additionally, The CASP14 gene exhibited relatively high expression in the skin, while its expression levels were low or even undetectable in the blood, central nervous system (CNS), heart, liver, lung and muscle.

3. Flow and Clarity:

The transitions between sections discussing CASP14's role in apoptosis and its relation to horn growth could benefit from smoother transitions to connect these ideas more explicitly.

More explanation is needed when introducing the Fst analysis results. The significance of Fst values should be briefly clarified for readers unfamiliar with population genetics.

Addressing these minor grammatical and technical inconsistencies would enhance the overall clarity and readability of the article while maintaining the strength of its scientific findings.

Author response: We agree with the reviewer’s assessments. We inserted the explanation on discussing the role of CASP14 in apoptosis and its relation to horn growth on line 240.

Line 240: The up-regulation of CASP14 by these lipids is mediated through the inhibition of kinase pathways, highlighting its role in cell differentiation and apoptosis [43]. Which may also impact the development of the horn phenotype.

In lines 103-105 of the manuscript, we have added a brief explanation of Fst value at its first mention to help clarify their significance for readers.

Lines 103-105: F-statistic (Fst) values were calculated using VCFtools (v.0.1.16) to identify SNP loci exhibiting significant differences between these two populations. Fst value is a measure of genetic differentiation between populations, with higher values indicating greater genetic divergence. 

We have addressed these minor issues to improve the article's clarity and readability while preserving the strength of our findings.

---

## [Editor Report · Decision Letter 1]

17 Oct 2024

Tissue-specific and functional loci analysis of CASP14 gene in the sheep horn

PONE-D-24-36938R1

Dear Dr. Pan,

We’re pleased to inform you that your manuscript has been judged scientifically suitable for publication and will be formally accepted for publication once it meets all outstanding technical requirements.

Kind regards,

Sayed Haidar Abbas Raza

Academic Editor

PLOS ONE

Additional Editor Comments (optional):

Thanks for respond the comments
---

## [Editor Report · Acceptance letter]

12 Nov 2024

PONE-D-24-36938R1 

PLOS ONE

Dear Dr. Pan, 

I'm pleased to inform you that your manuscript has been deemed suitable for publication in PLOS ONE. Congratulations! Your manuscript is now being handed over to our production team.

Kind regards, 

on behalf of

Dr. Sayed Haidar Abbas Raza 

Academic Editor

PLOS ONE